# Computationally efficient model of myocardial electromechanics for multiscale simulations

**Fyodor Syomin****\***, **Anna Osepyan, Andrey Tsaturyan**

Institute of Mechanics, Lomonosov Moscow State University, Moscow, Russia

\* f.syomin@imec.msu.ru

## Abstract

A model of myocardial electromechanics is suggested. It combines modified and simplified versions of previously published models of cardiac electrophysiology, excitation-contraction coupling, and mechanics. The mechano-calcium and mechano-electrical feedbacks, including the strain-dependence of the propagation velocity of the action potential, are also accounted for. The model reproduces changes in the twitch amplitude and $Ca^{2+}$-transients upon changes in muscle strain including the slow response. The model also reproduces the Bowditch effect and changes in the twitch amplitude and duration upon changes in the inter-stimulus interval, including accelerated relaxation at high stimulation frequency. Special efforts were taken to reduce the stiffness of the differential equations of the model. As a result, the equations can be integrated numerically with a relatively high time step making the model suitable for multiscale simulation of the human heart and allowing one to study the impact of myocardial mechanics on arrhythmias.

## Introduction

Mathematical modelling of myocardial electromechanics provides a useful tool for understanding processes of excitation, propagation of action potential (AP), $Ca^{2+}$-activation, and contraction in the normal and diseased heart. The modelling approach is particularly important because apart from direct excitation-contraction coupling, there are mechano-electrical and mechano-calcium feedbacks that make the system particularly complex for experimental studying. Detailed electrophysiological models of human ventricular or atrial cardiomyocytes and their clinical applications were reviewed recently [1, 2]. Some of these cell-level models [3–6] reproduce accurately the time-courses of various ionic currents, $Ca^{2+}$ transients, and the dependency of the AP duration and amplitude on the stimulation frequency. Such models allowed their authors to simulate various arrhythmias in the human heart and their drug or surgical treatment [1, 2, 7, 8]. One can combine electrophysiological models with a description of myocardial mechanics including both passive elastic and active $Ca^{2+}$-dependent stress. The electromechanical models, for example [9–11], allow one to examine numerically the effects of some molecular processes on disorders in muscle contraction or its excitation and can be incorporated into multiscale models of the human cardiovascular system for the simulation of

20-74-00046 to F.S. The funders had no role in study design, data collection and analysis, decision to publish, or preparation of the manuscript.

**Competing interests:** The authors have declared that no competing interests exist.

the whole heart or its chambers contraction. The combined models can be used to simulate the pumping function of the heart at different conditions [12] as well as they can be applied to the numerical investigation of electrophysiological activity of the heart [13].

However, the authors of the combined electromechanical models usually did not show any results relating to the influence of electrical stimulation on myocardial mechanics. It is unknown whether the models are capable to reproduce important dependencies of the twitch force on the stimulation frequency, including Bowditch effects and the force response to the variation in the interval between stimuli. The effect of the strain of myocardial tissue on the AP propagation velocity, so-called mechano-electrical feedback, was also not taken into account in most of those models with a few exceptions. The attempts of the modelling of the strong electromechanical coupling including the mechano-electrical feedback were made by a number of research groups [11, 14–19], however, the assumptions used in these models were not always supported by experimental observations, or only the mechano-calcium feedback was considered while a direct influence of strain on the speed of activation propagation was not taken into account. Another important feature of the force-frequency relation in cardiac muscle that was not demonstrated by previous simulations is the acceleration of force relaxation at increased stimulation frequency [20]. This acceleration is an important feature of the cardiac muscle as it provides sufficient time for refilling heart chambers even during shortened diastole at high heartbeat frequency.

Electromechanical models with a detailed description of ionic currents are usually expensive computationally as they require a very small time-step for numerical integration of kinetic equations. In contrast, phenomenological models of cardiomyocyte electrophysiology such as the Aliev-Panfilov model [21] are less demanding computationally although allow one to reproduce the AP propagation during normal heartbeats and arrhythmias [22–24].

Here we present a model that combines the Aliev-Panfilov model [21] with an improved version of our model of myocardial mechanics [25] extended by the introduction of major calcium currents and the additional balance equations for the $Ca^{2+}$ concentrations in the cytosol and sarcoplasmic reticulum (SR) based on the approach suggested in [3]. The mechanical model describes various mechanical phenomena including steady-state force-velocity and stiffness-velocity relations, tension responses to strain or load changes, tension redevelopment, load-dependent relaxation, etc. [25]. Special efforts were made to simplify the equations and make them less stiff. Our model of myocardial mechanics and its regulation by $Ca^{2+}$ ions was also modified to reduce computational time and to account for new structural data concerning the interaction of regulatory and contractile proteins. The equations of $Ca^{2+}$ balance from the model suggested by ten Tusscher and Panfilov were also simplified using an asymptotic approach to make them less stiff. Although the simplistic electrophysiology model cannot reproduce the time-courses of all ionic currents, the combined electromechanical model reproduces the $Ca^{2+}$-transients in different modes of cardiac muscle contraction. The model considers both mechano-calcium and mechano-electrical feedbacks through the effect of muscle strain on the $Ca^{2+}$-$Na^{+}$ exchange (NCX) and the conductivity and capacitance of cell membranes, respectively.

Our electromechanical model reproduces the effects of the stimulation frequency and interval between stimuli on the twitch force and intracellular free $Ca^{2+}$ concentration as well as instantaneous and slow response of twitch force to muscle stretch. Together with its tolerance for relatively high time step for numerical integration, this makes the model suitable for multiscale simulation of the heart including studying the strain-dependence of arrhythmias when the strain and interstimulus interval vary in space and time.

## Materials and methods

## Model description

**Electrophysiology and mechano-electrical feedback.**   The Aliev-Panfilov model [21] was used for the simulation of the membrane potential variation

$$\tau \frac{\partial(uC(\lambda))}{\partial t} = \nabla_j(D^{ij}\nabla_i u) - ku(u-a)(u-1) - uv + I_{stim},$$

$$\tau \frac{\partial v}{\partial t} = \left(\varepsilon + \frac{\mu_1 v}{\mu_2 + u}\right)(-v - ku(u-a-1)), \tag{1}$$

where $u$ is the dimensionless membrane potential related to the potential $V$: $V = 100u - °80$ [$mV$]; $v$ is a phenomenological dimensionless variable that describes the kinetics of ionic currents, and $I_{stim}$ is a dimensionless stimulation current; $D^{ij}$ are the contravariant components of the conductivity tensor $\hat{D}$ (there is summation over indexes $i$ and $j$ in Eq (1)); $\nabla_j$ is a covariant derivative over $j$-th spatial coordinate in a Lagrangian (material) coordinate system. Values of time scale $\tau$ and dimensionless constant parameters $k$, $a$, $\varepsilon$, $\mu_1$, $\mu_2$ are listed in Table 1. In transversally isotropic media, tensor $\hat{D}$ has the following form

$$\hat{D} = d_0\hat{E} + (d_1 - d_0)\hat{B}, \tag{2}$$

where $\hat{E}$ is the unit tensor, $\hat{B} = \vec{l} \otimes \vec{l}$ is the anisotropy tensor, and $\vec{l}$ is the unit vector directed along muscle fibres in strained muscle; $d_0$ and $d_1$ are isotropic and anisotropic conductivities ($d_1$ is the longitudinal conductivity along the fibre direction, $d_0$ is the transversal conductivity).

In order to account for the slowdown of the action potential (AP) propagation upon muscle stretch [26–28], the normalized capacitance of the cell membranes $C$ was assumed being strain-dependent [19]. The strain-dependence was shown to be associated with a strain-dependent recruitment of caveolae into the cell membrane causing an increase in its capacitance of and the electrical time constant even when stretch-activated ionic channels were inactivated [27]. Following [19] we assumed that the cell membrane capacitance $C$ normalized for its value at $\lambda = 1$ and $d_1$ depend on the axial strain $\lambda = \frac{l_s}{l_{s0}}$ as follows:

$$C = \begin{cases} 1 + \Delta_C \dfrac{(\lambda - 1)^{n_c}}{K_C{}^{n_c} + (\lambda - 1)^{n_c}}, \lambda > 1 \\ \qquad\qquad 1, \lambda \leq 1 \end{cases} \tag{3}$$

$$d_1 = d_1^0 \begin{cases} \dfrac{1}{1 + \Delta_d \dfrac{(\lambda - 1)^{n_d}}{K_d{}^{n_d} + (\lambda - 1)^{n_d}}}, \lambda > 1 \\ \qquad\qquad 1, \lambda \leq 1 \end{cases} \tag{4}$$

Here the values of the constant parameters: $\Delta_C$, $n_C$, $K_C$, $\Delta_d$, $n_d$, $K_d$ are the same as suggested by de Oliveira et al. [19]. The values of the parameters are given in Table 1.

In reality, the caveolae dynamics that controls the changes in the capacity and conductance are not instantaneous processes. Although the time-course of this process was not measured experimentally yet, it seems reasonable to assume that changes in $C$ and $d_1$ caused by strain are significantly slower than the change in membrane potential during onset and decay of AP in

**Table 1. The list of the model parameters.**

| Parameter | Meaning | Value | Units |
|---|---|---|---|
| $\alpha_{cyt}$ | Relative volume of cytosol | 0.75 | |
| $\alpha_{SR}$ | Relative volume of SR | 0.05 | |
| $a$ | Parameter of the A-P model | 0.1 | |
| $k$ | Parameter of the A-P model | 8 | |
| $\varepsilon$ | Parameter of the A-P model | 0.01 | |
| $\mu_1$ | Parameter of the A-P model | 0.2 | |
| $\mu_2$ | Parameter of the A-P model | 0.3 | |
| $\tau$ | Time scale of the A-P model | 12.9 | ms |
| $\Delta_C$ | Amplitude of strain dependence of membrane capacitance | 1 | |
| $n_c$ | Exponent for strain dependence of membrane capacitance | 6 | |
| $K_C$ | Parameter of strain dependence of membrane capacitance | 0.04 | |
| $\Delta_d$ | Amplitude of strain dependence of tissue conductivity | 0.4 | |
| $n_d$ | Exponent for strain dependence of tissue conductivity | 1 | |
| $K_d$ | Parameter of strain dependence of tissue conductivity | 0.02 | |
| $G_l$ | Rate constant of SR leakage | 0.03 | $S^{-1}$ |
| $l_{s0}$ | Reference sarcomere length | 1.9 | μm |
| $G_{CaL}$ | Rate constant of L-type $Ca^{2+}$ channels | 40 | $s^{-1}$ |
| $u_0$ | Constant parameter | 0.75 | |
| $k_u$ | Constant parameter | 13.33 | |
| $p_0$ | Parameter of polynomial approximation | 6.211 | |
| $p_1$ | Parameter of polynomial approximation | −7.233 | |
| $p_2$ | Parameter of polynomial approximation | 1.648 | |
| $G_{NCX}$ | Rate constant of $Na^+$-$Ca^{2+}$ exchange | 52500 | $s^{-1}$ |
| $Na_o$ | Extracellular $Na^+$ concentration | 140 | mM |
| $Na_{i0}$ | Reference intracellular $Na^+$ concentration | 10.5 | mM |
| $Ca_o$ | Extracellular $Ca^{2+}$ concentration | 2000 | mM |
| $K_{mNa}$ | Parameter of $Na^+$-$Ca^{2+}$ exchange | 87.5 | mM |
| $K_{mCa}$ | Parameter of $Na^+$-$Ca^{2+}$ exchange | 1300 | mM |
| $\eta$ | Parameter of $Na^+$-$Ca^{2+}$ exchange | 0.35 | |
| $\alpha$ | Parameter of $Na^+$-$Ca^{2+}$ exchange | 2.5 | |
| $k_{sat}$ | Parameter of $Na^+$-$Ca^{2+}$ exchange | 0.1 | |
| $k_{NCX}$ | SL sensitivity of $Na_i$ | 0.9 | |
| $G_{up}$ | Maximal rate of $Ca^{2+}$ uptake to SR | 500 | $s^{-1}$ |
| $K_{up}$ | Equilibrium constant for $Ca^{2+}$ uptake to SR | 0.4 | μM |
| $k_p$ | Rate constant of phosphlamban phosphorylation | 1 | $\mu M^{-2}s^{-1}$ |
| $K_p$ | Equilibrium constant of phosphlamban phosphorylation | 0.325 | μM |
| $G_{rel}$ | Rate constant of CICR | 150 | $s^{-1}$ |
| $K_{rel}$ | Equilibrium constant for CIRC | 400 | μM |
| $B_{cyt}$ | Buffer capacity for cytosol | 150 | μM |
| $K_{cyt}$ | Buffer affinity in cytosol | 1 | μM |
| $B_{SR}$ | Buffer capacity for SR | 10 | mM |
| $K_{SR}$ | Buffer affinity in SR | 1 | mM |
| $k_4$ | Rate constant for CIRC inactivation | 0.5 | $s^{-1}$ |
| $K_2$ | $Ca^{2+}$ affinity for CIRC inactivation | 0.5 | $\mu M^{-1}$ |
| $K_A$ | $Ca^{2+}$ binding constant for of activated TnC | 0.2 | μM |
| $K_B$ | $Ca^{2+}$ binding constant for blocked TnC | 2.5 | μM |
| $k_A$ | Rate constant for Tn activation | 50 | $s^{-1}$ |

(*Continued*)

**Table 1.** (Continued)

| Parameter | | Meaning | Value | Units |
|---|---|---|---|---|
| $\xi$ | | Parameter of Ca-TnC binding cooperativity | 0.35 | |
| $\kappa$ | | Parameter of Ca-TnC binding cooperativity | 3 | |
| | $k_{n1}$ | Parameter of Ca-TnC binding cooperativity | 40 | |
| | $k_l$ | Parameter of Ca-TnC binding length-dependence | 4 | |
| | $C_{Tn}$ | Total Tn concentration | 70 | μM |
| | $k_{01}$ | Rate constant of cross-bridge binding | 75 | s$^{-1}$ |
| | $k_P$ | Rate constant for phospholamban phosphorylation | 1 | s$^{-1}$ |
| | $K_P$ | Binding constant for phospholamban phosphorylation | 0.325 | μM |
| | $b$ | Parameter of the kinetic model of muscle mechanics | 1.5 | |
| | $c$ | Parameter of the kinetic model of muscle mechanics | 8.5 | |
| $\delta_*$ | | Parameter of the kinetic model of muscle mechanics | 0.3 | |
| $\delta_0$ | | Parameter of the kinetic model of muscle mechanics | 0.4 | |
| | $h$ | Cross-bridge step size | 10 | nm |
| | $E$ | Cross-bridge stiffness | 2.5 | pN/nm |
| $N_M$ | | A number of myosin filaments per a cross-section area of a sarcomere | $2.5\times10^{14}$ | m$^{-2}$ |
| $N_{xb}$ | | A number of myosin heads per a half of a myosin filament | 300 | |
| | $l_{s0}$ | Unstrained sarcomere length | 1.9 | μm |

A-P model refers to the Aliev-Panfilov model [21].

cardiac muscle cells. Therefore, the first equation in Eq (1) can be rewritten as

$$\tau C(\lambda)\frac{\partial u}{\partial t} = \nabla_j(D^{ij}\nabla_i u) - ku(u-a)(u-1) - uv + I_{stim}. \tag{1A}$$

**Calcium transport across cell membrane, mechano-calcium feedback.** Only two major components of Ca$^{2+}$ transport across the membrane of cardiomyocytes were taken into consideration: current through L-type Ca$^{2+}$-channels and calcium-sodium exchange, $I_{CaL}$, and $I_{NCX}$, respectively. We neglected any fast and slow gating of the L-type Ca$^{2+}$-channels and simply assumed the flux to be a function of the membrane potential $u$:

$$I_{CaL} = G_{CaL}\varphi(u), \varphi(u) = \frac{k_{CaL}(u-u)}{(1 + exp(k_u(u_0 - u)))(exp(k_{CaL}(u-u)) - 1)}, \tag{5}$$

where $G_{CaL}, u_0, u^*, k_u$, are constants specified in Table 1. To simplify calculations, the term $\frac{k_{CaL}(u-u)}{(exp(k_{CaL}(u-u))-1)} = p_0 + p_1 u + p_2 u^2$ in Eq (5) with nominator and denominator approaching zero at $u$ approaching $u^*$ was approximated with a second-order polynomial with an accuracy of 3%. Coefficients of the polynomial $p_i$ are also listed in Table 1.

The NCX was defined as suggested by ten Tusscher and Panfilov [3]

$$I_{NCX} = G_{NCX}\frac{[Na^+]_i^3[Ca^{2+}]_o exp\left(\eta\frac{Ve}{k_BT}\right) - \alpha c[Na^+]_o^3 exp\left((\eta - 1)\frac{Ve}{k_BT}\right)}{1 + k_{sat}exp\left((\eta - 1)\frac{Ve}{k_BT}\right)\left([Na^+]_o^3 + K_{Na}^3\right)\left([Ca^{2+}]_o + K_{Ca}\right)}, \tag{6}$$

where $Na_o = [Na^+]_o$, $Ca_o = [Ca^{2+}]_o$, $K_{Na}$, $K_{Ca}$ are constant extracellular concentrations of Na$^+$ and Ca$^{2+}$ and binding constants for Na$^+$ and Ca$^{2+}$ specified in Table 1; $c$ is Ca$^{2+}$ concentration in the cytoplasm, $V$ is the membrane potential (depending on $u$: $V = -80+100u$ [in mV]); $e$, $k_B$, $T$ are the elementary charge, the Boltzmann constant, and absolute temperature; $k_{sat}$, $\eta$, $\alpha$ are constant parameters from [3].

In our phenomenological model, the sodium concentration in the cytoplasm was not a variable that obeys a detailed equation for Na$^+$ balance as in [3, 29]. Instead, it was set as a function of sarcomere length.

$$[Na^+]_i = [Na^+]_{i0}(1 + k_{NCX}(\lambda - 1)), \tag{6A}$$

where $Na_{i0} = [Na^+]_{i0}$ and $k_{NCX}$ are constants specified in Table 1. Such a simple assumption within the framework of a phenomenological model is in line with the results of more detailed modelling of the slow response [30]. The slow response is a graduate increase in twitch amplitude upon a stretch of a cardiac muscle. These and other authors concluded that that the strain dependence of Na$^+$-H$^+$ exchange is a major contributor to the slow response [31, 32]. We show here that the simplest assumption (6A) is sufficient for explaining the amplitude and the time course of the slow responses of the twitch amplitude and Ca$^{2+}$ transients to an increase in muscle preload.

**Calcium transport into and from SR.** Ca$^{2+}$ uptake $I_{up}$ from the cytoplasm to the sarcoplasmic reticulum (SR) was defined similarly to that in [3, 33]:

$$I_{up} = G_{up}\frac{c^2 p}{K_{up}^2 + c^2}, \tag{7}$$

where $G_{up}$, $K_{up}$ are constants specified in Table 1. We have added a variable factor $p$ that is the level of phosphorylation of protein(s) controlling the Ca$^{2+}$ uptake to SR by sarcoplasmic reticulum Ca$^{2+}$-ATPase (SERCA) to account for the acceleration of muscle relaxation at high heartbeat rates.

There are several possible candidate mechanisms of posttranslational modification and target proteins responsible for the acceleration of the Ca$^{2+}$ uptake to SR upon an increase in the heartbeat rate. Phosphorylation of a small protein phospholamban (PLN) associated with Ca$^{2+}$ pump SERCA was the first candidate for frequency-dependent muscle relaxation [34]. Dephosphorylated PLN was shown to inhibit SERCA while its phosphorylation loses the inhibition so that the rate of Ca$^{2+}$ uptake to SR increases. Two PLN residues, Ser16 and Ser17, are phosphorylated by different kinases. The second one, Ser17 is phosphorylated by calmodulin-dependent kinase II (CaMKII) enhanced by high stimulation frequency [35]. Calmodulin is a Ca$^{2+}$ binding protein, therefore, the level of CaMKII phosphorylation should increase with an increase in time-average Ca$^{2+}$ concentration in cytosol. Later, some CaMKII-dependent acceleration of cardiac muscle relaxation was found in PLN-knockout animals [36] showing that other CaMKII-phosphorylated protein(s) including SERCA itself might be involved in the frequency dependence of cardiac muscle relaxation.

Although the precise mechanism of accelerated relaxation at high stimulation rate is still debated [37, 38], the presence of such mechanism, which most probably involves CaMKII, is well established. We do not specify phosphorylation of which particular protein(s) is responsible for the acceleration of the Ca$^{2+}$ uptake into SR in our phenomenological model. Instead, we simply specify the dependence of its phosphorylation level p on cytoplasmic Ca$^{2+}$ concentrations $c$ as follows:

$$\frac{\partial p}{\partial t} = k_p\left(c^2(1 - p) - K_p^2 p\right), \tag{8}$$

where $k_p$, $K_p$ are constants characterizing the phosphorylation rate and the equilibrium constant, respectively; values of both parameters are presented in Table 1.

The major source of Ca$^2$ in the cytosol is calcium-induced calcium release (CICR) from SR. CICR is a complex process activated by increased Ca$^{2+}$ concentration in the narrow subspace between the cell membrane with the L-type Ca$^{2+}$-channels (dihydropyridine receptors; DHPR) and the SR membrane with ryanodine receptors (RyR) as well as by high Ca$^{2+}$ concentration in the SR lumen. Several sophisticated models partially reviewed by Hinch et al. [39] as well as a non-Markovian one [40] and a model of interacting clusters of DHPR and RyR [41] were suggested for simulation of different aspects of CICR. Here we neglected all details that are not necessary for describing major feature of CIRC in normally contracting cardiac muscle cell and set the CICR flux in a rather simple form [3]:

$$I_{rel} = G_{rel} R (c_{SR} - c_{SS}) \frac{c_{SS}^2}{c_{SS}^2 + K_{rel}^2},$$ (9)

The kinetics of closing gate $R$ was described by the equation

$$\frac{\partial R}{\partial t} = -k_4 \left( K_2 \frac{c_{SS} R}{C_{SS} + K_R} + R - 1 \right).$$ (10)

Here $G_{rel}$, $K_{rel}$, $K_2$, $k_4$, $K_R$ are constants specified in Table 1. Eqs (9–10) are based on the assumption that CIRC activation is very fast although having relatively low sensitivity for Ca$^{2+}$ concentration in the narrow subspace between L-type Ca$^{2+}$ channels and RyR, $c_{SS}$, while CIRC deactivation is much slower although more sensitive to the Ca$^{2+}$ concentration in the subspace. In contrast to [3] we assume that neither the CIRC activation nor its inactivation is sensitive to the amount of Ca$^{2+}$ in SR as the model was able to describe the frequency-dependence of twitch amplitude without such assumption (see below). Besides, we have modified Eq (10) by taking into account a relatively high sensitivity of RyR to the Ca$^{2+}$ concentration in subspace $c_{ss}$.

**Calcium-binding to Tn and kinetics of Tpm-Tn regulation.** Muscle contraction is activated by binding of Ca$^{2+}$ ions to troponin C (TnC), a subunit of troponin (Tn) complex of three regulatory proteins. The Ca$^{2+}$ binding leads to conformation changes causing a shift in the position of another regulatory protein tropomyosin (Tpm), which blocks the binding sites for myosin on actin in the absence of Ca$^{2+}$ and opens them upon Ca$^{2+}$ binding to TnC. Following Fusi *et al.* [42] who have shown that Ca$^{2+}$ binding to TnC is fast, we assumed that the Ca$^{2+}$ binding to TnC in the blocked or activated (closed plus open) states of the regulatory units of the thin filaments is quick and reversible although the affinities of TnC for Ca$^{2+}$ are different. Accordingly, the fractions of Ca$^{2+}$-bound TnC molecules in the activated and blocked regulatory units were set as follows:

$$[CaTn]_{Bi} = \frac{c(1 - A_i)}{c + K_B}, \; [CaTn]_{Ai} = \frac{cA_i}{c + K_A},$$ (11)

where $B$ and $A$ refer to the blocked and activated states of the regulatory units, respectively; index $i$ = 1, 2 corresponds to the overlap and non-overlap zones of the thin and thick filaments in sarcomeres, respectively. The activated state is a combination of the closed and open states of the Tn-Tpm complex according to the three-state model of McKillop, Geeves [43]. The equilibrium constants $K_A$, $K_B$ (given in Table 1) are different as TnC in the activated regulatory units bind Ca$^{2+}$ tighter than in the blocked ones [44]. The rate-limiting step of the Ca$^{2+}$ activation of a striated muscle is the transition from the blocked to activated state [42]. The transition requires the detachment of the inhibitory domain of the troponin subunit troponin I (TnI) from actin and Tpm and binding of TnI switch segment to the hydrophobic pocket on TnC that opens upon Ca$^{2+}$ binding to TnC [45, 46].

The kinetics of activation of the regulatory units in the overlap $A_1$ and non-overlap $A_2$ zones of the thin filaments was described by equations:

$$\frac{\partial A_i}{\partial t} = k_A \left( \frac{c(1 - A_i)}{c + K_B}(1 + k_l(\lambda - 1)) - A_i \frac{K_A(1 - \xi + \xi exp(k_{ni} n \theta(\delta))) exp(\kappa(A_i - 0.5))}{c + K_A} \right) +$$

$$\begin{cases} 0, \text{ if } \dfrac{\partial W(\lambda)}{\partial t} \leq 0, i = 1 \vee \dfrac{\partial W(\lambda)}{\partial t} \geq 0, i = 2 \\[2mm] \dfrac{\partial W(\lambda)}{\partial t} \dfrac{(A_2 - A_1)}{W(\lambda)}, i = 1, \dfrac{\partial W(\lambda)}{\partial t} > 0 \\[2mm] \dfrac{\partial W(\lambda)}{\partial t} \dfrac{W_{max}(A_2 - A_1)}{(l_a - W_{max} W(\lambda))}, i = 2, \dfrac{\partial W(\lambda)}{\partial t} < 0 \end{cases} \quad , \tag{12}$$

Where constant parameters $k_A$, $\kappa$, $\xi$, $k_l$, $k_{n1}$ ($k_{n2} = 0$) and the length of an actin filament in a sarcomere $l_a$ are specified in Table 1. $W$, $W_{max} = 0.5(l_m - l_b)$ are the normalized length of the overlap zone of the thin and thick filaments in sarcomere and the maximal length of this zone; $l_m$ and $l_b$ are the length of the thick filament and its bare zone. These equations are similar although different from those in [25]. Parameters $\kappa$, $\xi$ characterize cooperativity between activation of neighbor regulatory units. TnC binds $Ca^{2+}$ in both the blocked and activated regulatory units although with different affinities as suggested by [44]. In addition, the Eq (12) is less stiff than that used in [25] and therefore more convenient from the computational point of view. It should be noted that the last "convective" term in Eq (12) can be omitted if the strain rate of sarcomere normalized for the sarcomere length is small compared to the rate constant of $Ca^{2+}$ activation $k_A$. Eq (12) then reduces to

$$\frac{\partial A_i}{\partial t} = k_A \left( \frac{c(1 - A_i)}{c + K_B}(1 + k_l(\lambda - 1)) - A_i \frac{K_A(1 - \xi + \xi exp(k_{ni} n \theta(\delta))) exp(\kappa(A_i - 0.5))}{c + K_A} \right). \tag{13}$$

**Calcium-mechanical coupling.** We slightly modified the system of the equations for calcium-mechanical coupling based on our model of cardiac cell mechanics [25]. The mechanical model described the kinetics of generation of active tension $T_a$ by contractile proteins myosin and actin and $Ca^{2+}$ activation of their interaction. We modified the model equations to reduce the stiffness of the system and to describe the force-$Ca^{2+}$ relationships more accurately. The expression for active tension was the same as in the previous model.

$$T_a = EN_M N_{xb} n \cdot W(\lambda)(\delta + \theta(\delta)), \tag{14}$$

while the equations for the kinetics of the interaction of the contractile proteins were slightly modified:

$$\frac{\partial n}{\partial t} = k_{01}\left(F(\delta)(A_1^2 - n) - nG(\delta)\right), \tag{15}$$

$$\frac{\partial \delta}{\partial t} = \begin{cases} D, D < 0 \vee \delta \leq \delta_* \\ 0, \text{ otherwise,} \end{cases} \tag{16}$$

where

$$D = \frac{l_{s0}}{2h}\frac{\partial \lambda}{\partial t} - \delta F(\delta)\frac{A_1^2 - n}{n}.$$

Here $n$, $\delta$, $A_1$ are the fraction of the actin-bound myosin heads, ensemble-averaged distortion of actin-bound heads per sarcomere, and the normalized activation of regulatory tropomyosin-troponin (Tpm-Tn) strand in the overlap zone of sarcomeres, respectively. Functions $F(\delta)$, $G(\delta)$, $W(\lambda)$, $\theta(\delta)$ and constants $E$, $N_M$, $N_{xb}$, $h$, $k_{01}$, $l_{s0}$ were the same as suggested previously in [25], and new constant $\delta^*$ is specified in Table 1. From the mechanical point of view, the threshold $\delta^*$ defines a transition from viscoelasticity to plasticity at high stretch rate and reduces the stiffness of the differential equations at a high stretch rate. One should note that the binding of myosin heads in Eq (15) is controlled by $A_1$ squared instead of $A_1$ itself as it was in [25]. This assumption was motivated by recent structural data showing an interaction between two Tpm-Tn strands *via* the N-terminal part of troponin T (TnT) that binds the Tpm strand at the region of the overlap junction of two consecutive Tpm dimers arranged on the opposite sides of the thin filament [46]. Such interaction should induce cooperativity of the azimuthal movement of the two Tpm strands so that two regulatory units on both sides of the thin filament undergo the block-to-open transition together when both TnI on the opposite sides of an actin filament are released from actin.

**Calcium balance.** The calcium concentration in subspace $c_{SS}$ obeys the balance equation from [3]

$$\alpha_{ss}\frac{\partial}{\partial t}\left(c_{SS} + \frac{c_{SS}B_{ss}}{c_{SS} + K_{SS}}\right) = I_{rel} + I_{CaL} - G_{xfer}(c_{SS} - c), \tag{17}$$

where $\alpha_{ss} \ll 1$, $B_{ss}$, $K_{ss}$ are relative volume of subspace, $Ca^{2+}$ buffer concentration in subspace, and its equilibrium constant; $G_{xfer}$ is the rate constant of $Ca^{2+}$ diffusion from the subspace to the cytosol. The smallness of $\alpha_{ss}$ makes the system of differential equations stiff, so that it requires a small time step and\or special numerical methods for its integration. On the other hand, the smallness of $\alpha_{ss}$ allows one to substitute Eq (11) with an asymptotic solution at $\alpha_{ss} \to 0$ using Tikhonov theorem [47], i.e., to set the right-hand side of Eq (17) to zero provided that the solution of the reduced equation is a stable root of Eq (18).

$$G_{xfer}(c_{SS} - c) - G_{rel}R(c_{SR} - c_{SS})\frac{c_{SS}^2}{c_{SS}^2 + K_{rel}^2} - G_{CaL}\varphi(u) = 0. \tag{18}$$

Eq (18) is a cubic equation for $c_{SS}$ at constant or slowly changing $c$, $c_{SR}$, and $u$:

$$(G_{rel}R + G_{xfer})c_{SS}^3 - (G_{rel}Rc_{SR} + G_{xfer}c + G_{CaL}\varphi(u))c_{SS}^2 + G_{xfer}K_{rel}^2c_{SS} - K_{rel}^2(G_{xfer}c + G_{CaL}\varphi(u))$$
$$= 0. \tag{19}$$

Its solution obtained by the Cardano formulas gives the explicit expression $c_{SS}(c_{SR}, c, R, u)$ that was substituted to Eqs (9, 10) and the equations describing $Ca^{2+}$ balance in cytosol and SR (see below). Depending on the model parameters, Eq (13) can have one, two or three real positive solutions. For the case of three solutions, the smallest and highest solutions are stable, while the intermediate one is unstable. More details of the choice of a solution of Eq (19) are given in S1 File.

Both cytosol and SR contain $Ca^{2+}$ buffers different from TnC with the capacities (total concentration) $B_{cyt}$ and $B_{sr}$, respectively, and the equilibrium constants $K_{cyt}$, $K_{sr}$, respectively (for values see Table 1). Both buffers were assumed to bind $Ca^{2+}$ very quickly and reversibly so that

the equation of $Ca^{2+}$ balance in cytosol was formulated as follows:

$$\alpha_{cyt}\frac{\partial}{\partial t}\left(c+\frac{B_{cyt}c}{c+K_{cyt}}\right)=I_{NCX}-I_{up}+G_{xfer}(c_{SS}-c)+G_{leak}(c_{SR}-c)-I_{Tn}, \qquad (20)$$

where $\alpha_{cyt}$ is a relative volume of cytosol; $G_{leak}$ is the coefficient that characterizes $Ca^{2+}$ leak from SR to the cytosol. The total concentration of $Ca^{2+}$ bound to TnC is given by expression.

$$C_{TnC}=\frac{C_{Tn}}{l_a}\left(\frac{C}{C+K_B}(W_{max}W(\lambda)(1-A_1)+(l_a-W_{max}W(\lambda))(1-A_2))+\frac{C}{C+K_A}(W_{max}W(l_s)A_1+(l_a-W_{max}W(l_s))A_2)\right). (21)$$

Therefore, the flux of free $Ca^{2+}$ from cytosol due to its binding to TnC $I_{Tn}$ is as follows:

$$\begin{aligned}I_{Tn}&=\frac{\partial C_{TnC}}{\partial t}\\&=\frac{C_{Tn}}{l_a}\frac{\partial}{\partial t}\left(\frac{C}{C+K_B}(W_{max}W(l_s)(1-A_1)+(l_a-W_{max}W(l_s))(1-A_2))+\frac{C}{C+K_A}(W_{max}W(l_s)A_1+(l_a-W_{max}W(l_s))A_2)\right),(22)\end{aligned}$$

where $C_{Tn}$, is the total concentration of Tn. Substituting Eq (22) into Eq (20) one can obtain the equation for $Ca^{2+}$ balance in the cytosol. $Ca^{2+}$ balance in SR was described by the equation.

$$\alpha_{sr}\frac{\partial}{\partial t}\left(c_{SR}+\frac{B_{sr}c_{SR}}{c_{SR}+K_{sr}}\right)=I_{up}-I_{rel}-G_{leak}(c_{SR}-c). \qquad (23)$$

Here $\alpha_{sr}$ are relative volumes of SR in cardiac muscle tissue, $B_{SR}$, $K_{SR}$ are the total concentration of $Ca^{2+}$ buffer in SR and its equilibrium constant, respectively.

## Results

For simulation of contraction of a single cardiac muscle cell or homogeneously stimulated cylindrical multi-cell samples, the model reduces to a 0D problem specified by the set of ordinary differential equations: Eq (1) without diffusion term as the membrane potential does not depend on coordinates, Eqs (8), (10), (15), (16), (20), (22), (23). The mechanical problem setup and the total tension specification are described in S1 File. The system of the equations was solved numerically together with mechanical equations defining the load or strain of cardiac muscle using forward Euler method with the time step of 0.1, 0.2 and 0.05 ms. The time step of 0.1 ms provided good convergence (see S3 Fig in S1 File). In the simplest case of 0D homogenous isometric contraction at a constant sarcomere length $l_s$, $\delta = 0$ and Eq (17) can be omitted. Isotropic and anisotropic (titin) components of passive elastic stress were the same as in [25].

### Steady-state $Ca^{2+}$ regulation, effect of sarcomere length

Steady-state dependencies of normalized active tension $T_{A,norm}$, the normalised fractions of activated Tn-Tpm regulatory units in overlap zone and outside it $A_1$, $A_2$, and normalized concentration of $Ca^{2+}$-bound Tn in the overlap and non-overlap zones of sarcomere at different constant sarcomere length ($[CaTn]_1$, $[CaTn]_2$) are shown in Fig 1. The active tension $T_a$ was normalized for its steady-state value at saturating $Ca^{2+}$ concentration at full overlap of the thin and thick filaments in sarcomeres. The $Ca^{2+}$-tension relation at different sarcomere length for the model (Fig 1(C), 1(D)) was similar to that observed experimentally in rat [48] and human [49] cardiac muscles. A decrease in the sarcomere length by 0.4 μm lead to a ~40% increase in apparent equilibrium constant estimated from normalized $Ca^{2+}$-force relation (Fig 1(D)). The $Ca^{2+}$-force relation obtained from the model is similar to that predicted by the Hill equation although is more steep at low $Ca^{2+}$ concentration and less steep at high $Ca^{2+}$ concentration as

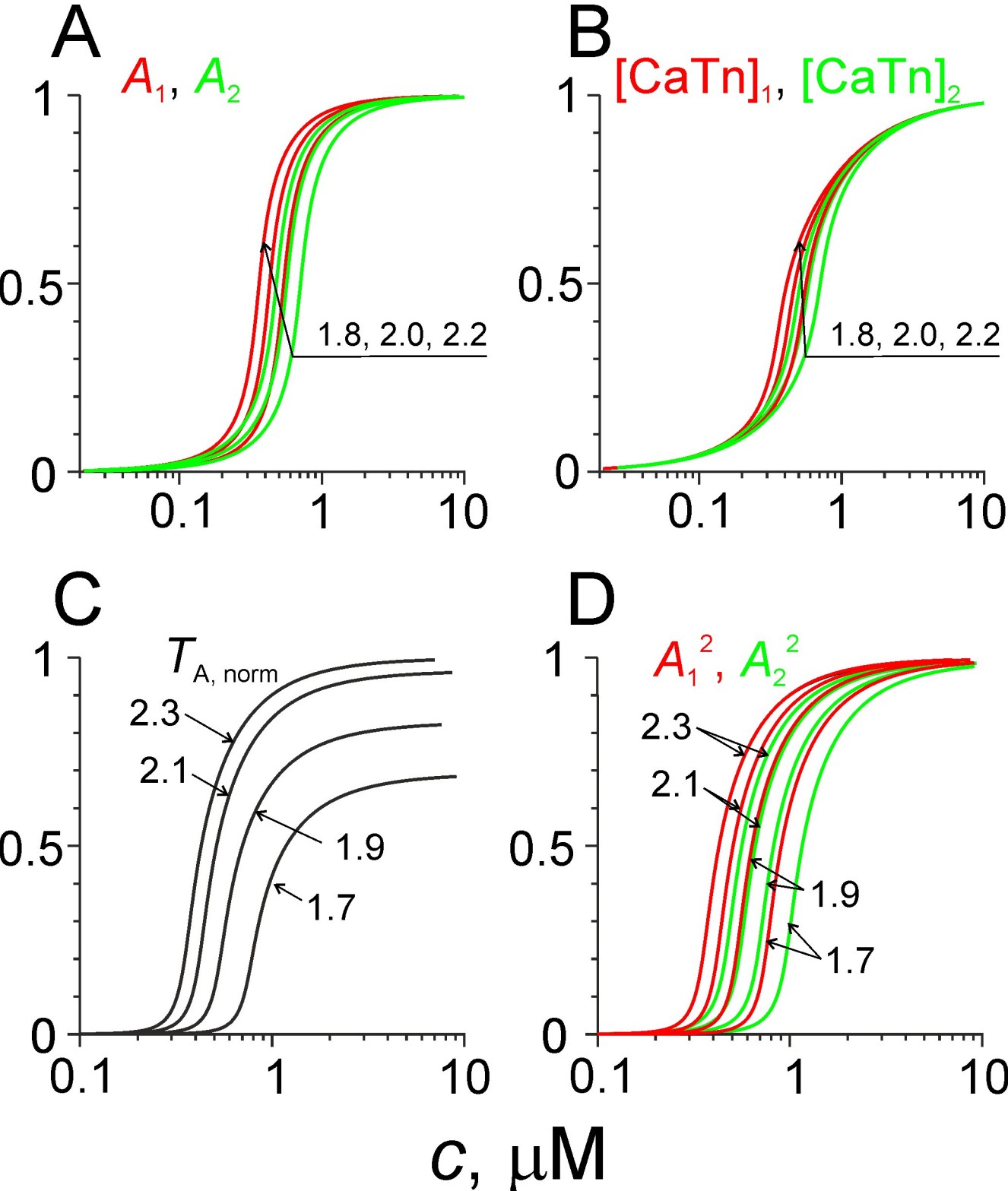

**Fig 1. Simulation of steady-state muscle contractions.** Calculated $Ca^{2+}$-dependencies of $A_1$ and $A_2$ (red and green, respectively, A), normalised CaTn concentration in the overlap (red) and non-overlap (green) zones of a sarcomere (B), normalized active tension (C) and $A_1^2$ and $A_2^2$ (D) at different sarcomere length are shown. The length values are shown next to the curves in μm.

found by Dobesh et al. in [48]. The shift in the $Ca^{2+}$ sensitivity caused by myosin cross-bridges can estimated from the difference between $A_1^2$ and $A_2^2$ in Fig 1(D). It was similar to that obtained in experiments with substances which inhibit myosin binding to actin [44]. The $Ca^{2+}$-sensitivity of active isometric tension for the model was higher than that observed in experiments with skinned muscle cells and multicellular specimens although was similar to those in intact trabeculae [50]. As the model was designed for describing intact human cardiac muscle, $Ca^{2+}$ binding constants for Tn were adjusted to fit available data obtained on intact human cardiac muscles.

### Time course of model variables during twitch contractions, effect of sarcomere length

The time course of all model variables and major $Ca^{2+}$ flows during an isometric twitch contraction at sarcomere length 2.2 μm at a stimulation rate 1 Hz are shown in Fig 2.

The time courses of tension and cytosol $Ca^{2+}$ concentration $c$ were similar to those observed experimentally in human cardiac muscle [51]. Sharp instantaneous changes in $c_{SS}$ (Fig 2(A)) were caused by jumps of the Cardano solution of the cubic Eq (19) when system passed a bifurcation point. The difference in the time course of $A_1$ and $A_2$ is induced by more tight $Ca^{2+}$ binding to Tn in the overlap zone due a cross-bridge Tmp-Tn interaction. Outward $Ca^{2+}$ flow thorough NCX during diastole balances its inward flow and flow through the L-type $Ca^{2+}$ channels during systole. The total amount of $Ca^{2+}$ entering cytosol from SR was about twice higher than that entering cell during systole via NCX and L-type channels (Fig 2(B)).

The effects of sarcomere length on the parameters of twitch contractions are shown in Fig 3.

The model describes an increase in the twitch amplitude and duration upon a muscle stretch. Apart from an instantaneous rise in twitch amplitude there was an additional ~25% increase that develops for several tens of seconds (Fig 3(A)). A slow response very similar to that shown in Fig 3 was observed in cardiac muscle samples from human [52] and other mammalians [53]. Just after stretch, calculated twitch amplitude increased, while the amplitude of

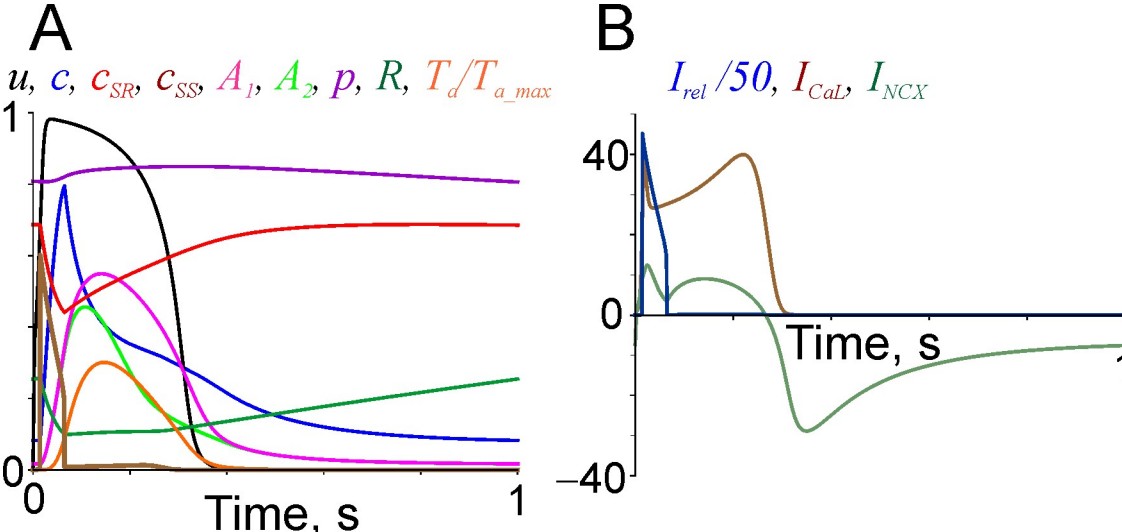

**Fig 2. The time courses of model variables and major $Ca^{2+}$ flows during a twitch contraction at constant sarcomere length of 2.2 μm and stimulation rate of 1 Hz.** A: $u$, $c$ (in μM), $c_{SR}$ and $c_{SS}$ (in mM), $A_1$, $A_2$, $R$, and $T_a/T_{a\_max}$ (active tension normalised for its maximal isometric value at saturating $Ca^{2+}$ concentration at the same sarcomere length); B: $I_{NCX}$, $I_{CaL}$, and $I_{rel}/50$ (in μM×s$^{-1}$).

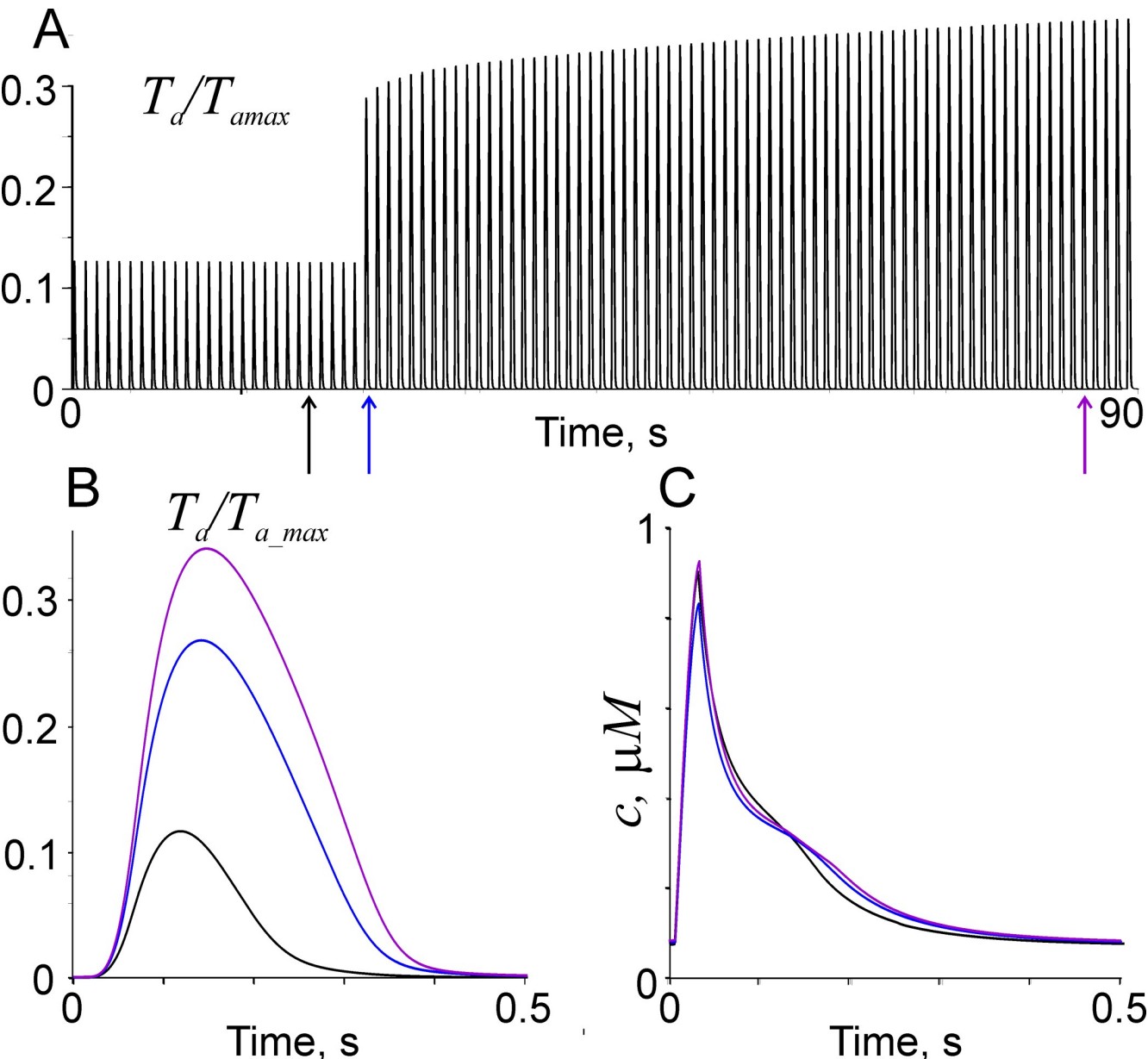

**Fig 3. The effect of sarcomere length on twitch contractions at the stimulation rate 1 Hz.** A: isometric twitches before and after a change of sarcomere length from 1.9 to 2.2 μm; B, C: calculated normalised active tension and $Ca^{2+}$ concentration in cytosol for twitches shown by arrows of the same colour in A.

$Ca^{2+}$ transient slightly decreased. During the slow response, the peak $Ca^{2+}$ gradually increases to its pre-stretch value. The fast decrease in $Ca^{2+}$ concentration just after the peak accelerates after the stretch. The slower decrease in the $Ca^{2+}$ concentration in cytosol after the peak of the twitch decelerates upon the stretch as observed experimentally [54, 55]. The graduate increase in the peak $Ca^{2+}$ concentration caused further increase in the twitch amplitude (Fig 3(B), 3(C)). These changes were caused by more tight $Ca^{2+}$ binding to TnC at longer sarcomere length. In our model, the slow response was caused by an increased $Ca^{2+}$ influx into cardiac muscle cells via NCX due to a strain-dependent increase in intracellular $Na^+$ concentration. As

the model does not include detailed description of $Na^+$ balance, there is a single parameter $k_{NCX}$ responsible for the slow response in our model.

## Effect of stimulation frequency and interval between stimuli on twitch amplitude and duration

Changes in twitch amplitude upon a change in the stimulation rate are shown in Fig 4.

The model reproduces so called Bowditch effect, i.e., an initial decrease in the twitch amplitude just after an increase in the stimulation rate followed by its graduate increase (Fig 4, top) and biphasic response to a decrease in the stimulation rate: an increase in the amplitude after first elongated interval and its graduate decrease in the course of subsequent low-frequency stimulation (Fig 4, bottom).

The dependence of the steady-state twitch amplitude on the stimulation frequency and on the duration of the interval after long time stimulation at a frequency of 1 Hz are shown in Fig 5. Two types of experiments were simulated: long-term periodical muscle stimulation at different constant frequency (Fig 5(A)) and the experiment when a stimulus is applied with different intervals after long-time stimulation at 1 Hz (Fig 5(B)).

The force-frequency dependence shown in Fig 5(A) is similar to that obtained for human cardiac muscle samples [20, 56, 57]. The frequency dependence of the peak $Ca^{2+}$ concentration (Fig 5(A)) was also similar to that observed in cardiac muscle samples from non-failing human hearts [51]. The increase in the twitch amplitude with an increase in the stimulation rate in our model was provided by two factors: i) an increase in the time-average $Ca^{2+}$ influx via the L-type $Ca^{2+}$ channels caused by an increase in the ratio of systole duration to the heartbeat circle duration that led to $Ca^{2+}$ accumulation in SR; and ii) an increase in $Ca^{2+}$ accumulation in SR due to increased $Ca^{2+}$ uptake rate by SERCA caused by a more active phosphorylation of phospholamban or other SERCA-associated proteins at higher average $Ca^{2+}$ concentration in

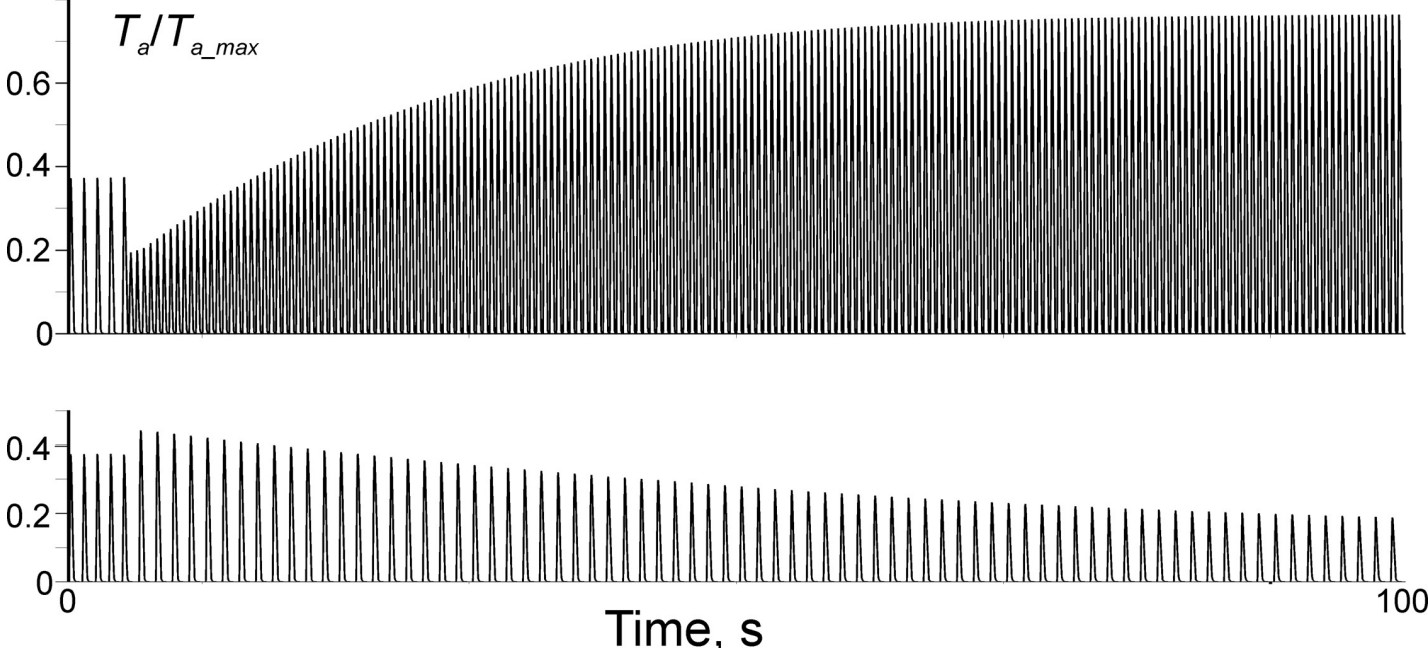

**Fig 4. Changes in twitch amplitude (normalised active tension) upon changes of stimulation frequency.** The stimulation frequency changed from 1 Hz to 2 Hz (upper trace) and from 1 Hz to 0.8 Hz (lower trace).

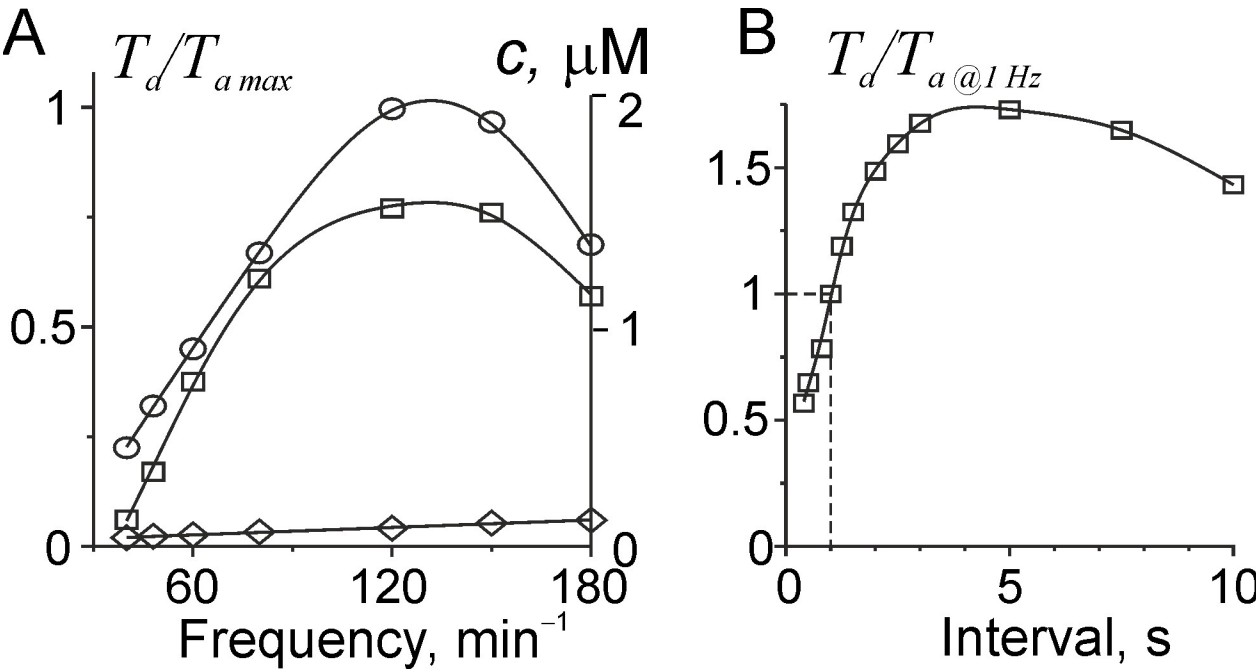

**Fig 5. Contraction dependence on the interstimulus interval.** A: the dependence of calculated active tension normalized for its steady-state value at saturating $Ca^{2+}$ concentration at the same sarcomere length (squares), maximal (circles) and minimal (diamonds) $Ca^{2+}$ concentration $c$ in cytosol (right scale) on the stimulation frequency. B: the dependence of peak twitch tension normalized for its value at 1 Hz stimulation on the interval duration after 1 Hz stimulation.

cytosol. The model also describes the effect of extra systole (for intervals less than 1 s) and post pause potentiation at interval longer than 1 s (Fig 5(B)).

The time courses of normalized active tension at different stimulation rates are shown in Fig 6.

The model describes a decrease in the twitch duration at higher stimulation rate observed experimentally in cardiac muscle specimens from different mammals including humans [20]. In our model, the acceleration of twitch relaxation with frequency is explained by an increase in the rate of $Ca^{2+}$ uptake into SR at higher time-averaged $Ca^{2+}$ concentration in cytosol due to the enhanced phosphorylation of SERCA-associated protein(s) defined by Eqs (7) and (8).

The results of simulation of isotonic twitch contraction at different load are shown in Fig 7. The lower was the load the higher was shortening amplitude. Importantly, shortening under low load accelerated muscle relaxation that became significantly faster than that for isometric contraction, showing so-called load-dependent relaxation of the left ventricular myocardium [58, 59].

## Discussion

### Aim of the work and its main results

The aim of the work was to build an electromechanical model suitable for multiscale simulation of the human heart. The main requirements for the model were:

- its ability to simulate changes in major electrophysiological parameters: duration and propagation velocity of AP as well as instantaneous and slow mechanical responses to strain;

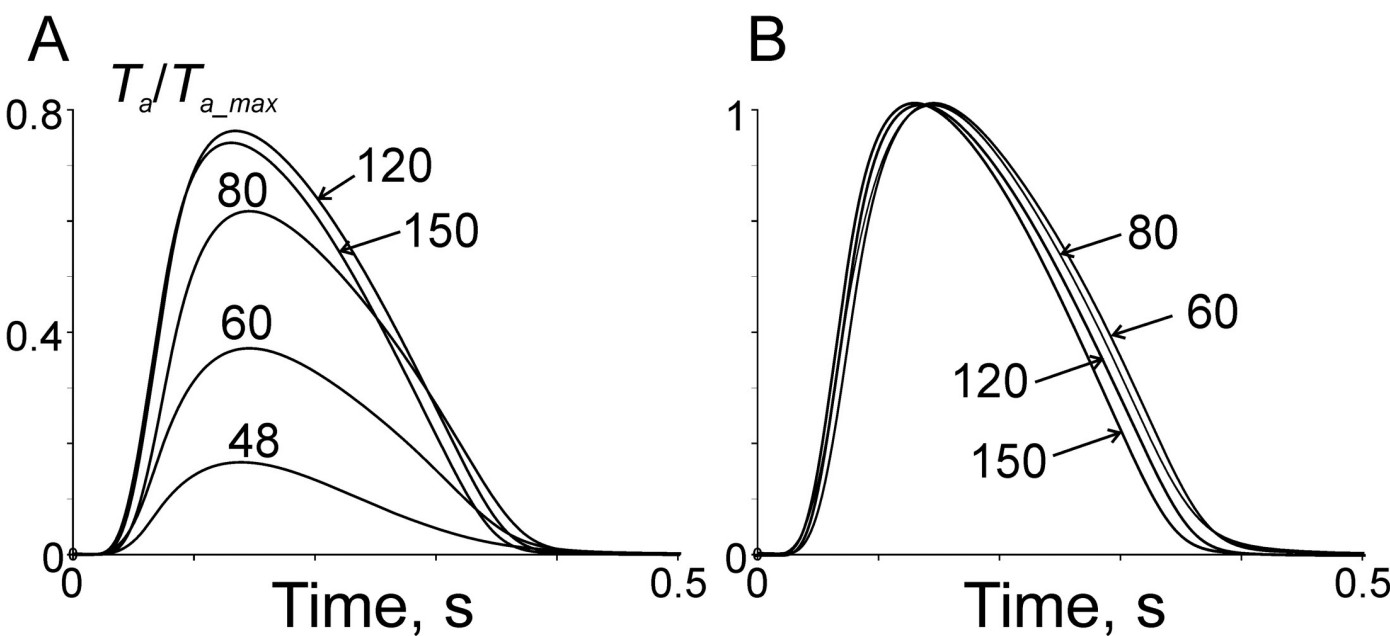

**Fig 6. Calculated active tension during isometric twitches at different stimulations rates at sarcomere length 2.2 μm.** The rates are shown next to curves in min$^{-1}$. A: active tension normalized for its maximal value at saturating Ca$^{2+}$ concentration; B: same calculated tension traces normalized for their peak values.

- the ability of the model to simulate instantaneous and long-term changes in twitch amplitude and duration caused by variation of the interval between subsequent APs;

- the ability of the model to describe all major mechanical properties of cardiac muscle such as force-velocity and stiffness-velocity relations, non-steady responses to changes in muscle length or load including load-dependent relaxation;

- the model should be relatively simple computationally for its usage in the multiscale simulations.

A number of models with detailed descriptions of various ion currents, AP generation, and propagation have been suggested [1–6]. Mechano-electrical and mechano-calcium feedbacks causing changes in the time course of AP and Ca$^{2+}$ transients and the twitch amplitude were also taken into account by some detailed electromechanical models in 0D simulations [10, 14, 30, 60]. A data-based model of the strain-dependence of the AP propagation velocity was suggested [19] although the possible role of this type of mechano-electrical feedback in arrhythmogenesis was not studied.

However, the electromechanical models with detailed descriptions of ion currents require a very small time-step and very fine space mesh for numerical simulation. This makes simulation of heart electromechanics with realistic geometry for several seconds prohibitory expensive computationally making the stimulation of several heartbeat cycles with varying intervals between them or during an arrhythmia with complex pattern of activation and contraction almost impossible. To avoid this problem, we decided to use a popular phenomenological model of cardiac electrophysiology [21] that describes the observed dependence of the AP duration on the stimulation frequency. The model was shown to be able to simulate various cardiac arrhythmias [15, 24, 61], The mechano-electrical feedback was simulated in our model according to de Oliveira et al. [19] with a slight modification to account for its anisotropy.

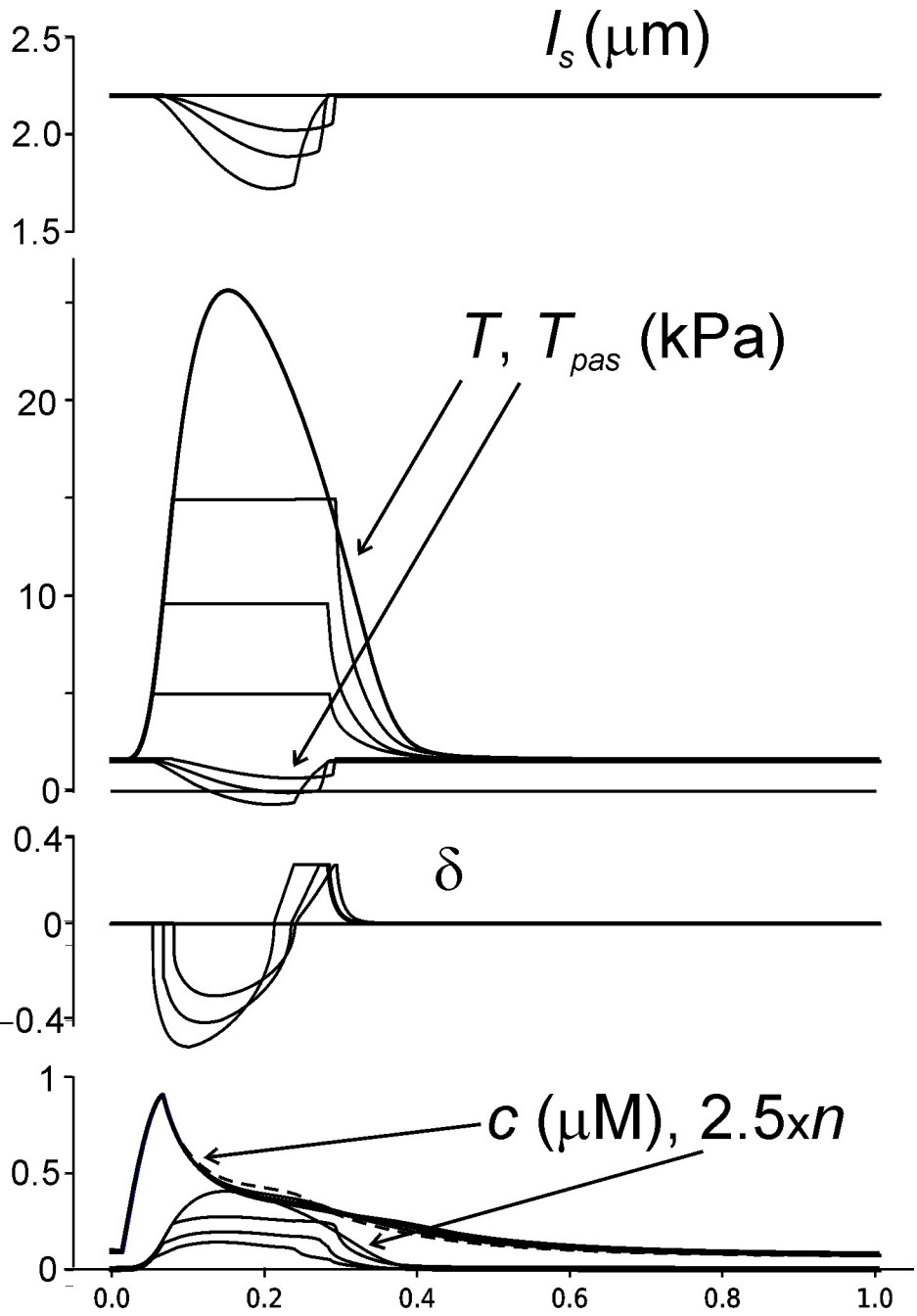

**Fig 7. The results of simulation of isotonic contractions under different load.** From top to bottom: sarcomere length, total and passive tension, $\delta$, $c$, and $n$ multiplied by a factor 2.5. $Ca^{2+}$-transient $c$ for isometric contraction is shown by dashed line.

Cardiac muscle mechanics, calcium-mechanical coupling, and mechano-calcium feedback were described using a modification of our model [25]. The modification accounts for new structural data concerning the molecular mechanics of $Ca^{2+}$ regulation of cardiac muscle contraction [46, 62] and, importantly, makes the system of equations that describes the $Ca^{2+}$-activation of the thin filaments less stiff. Besides, an equation that describes the kinetics of the

actin-myosin interaction was modified to avoid singularity at the high speed of muscle stretch. The model of electro-calcium coupling was adopted from [3] and simplified by neglecting some minor $Ca^{2+}$ currents and by using an asymptotic form of the description of $Ca^{2+}$ concentration in the narrow subspace between the L-type $Ca^{2+}$ channels, the T-tubule, and ryanodine receptors in the SR membrane. A novel aspect of the model is the dependence of the ATP-dependent $Ca^{2+}$ uptake into SR on the phosphorylation of protein(s) associated with SERCA that is believed to be responsible for the acceleration of muscle relaxation at high stimulation frequency [37].

## Estimation of parameters and verification of the model

We used standard values of the parameters of the Aliev-Panfilov model [21] in Eq (1) and the same parameters of the mechano-electrical feedback in Eqs (3) and (4) as those suggested by de Oliveira et al. [19]. The parameters of the model for the $Ca^{2+}$ flows between cytosol, SR, and subspace as well as the parameters of the $Ca^{2+}$ currents via L-type $Ca^{2+}$-channels in Eq (5) and $Na^+$-$Ca^{2+}$ exchange in Eq (6) were the same as in [3]. The parameters of the actin-myosin interaction were the same as in [25] with an additional parameter $\delta^*$ that describes the transition from viscoelasticity to plasticity upon muscle stretch at high velocity. The parameters of the calcium-mechanical coupling describing $Ca^{2+}$ binding to TnC in the blocked and activated states and the kinetics of the transition between these states (Eqs 11) and (13) were adjusted to fit the data on the $Ca^{2+}$-tension relations at different sarcomere length [48, 49] and the data on the effect of actin-bound myosin heads on the activation of the thin filaments [44]. A correction was made to account for the difference between $Ca^{2+}$ binding to Tn in skinned and intact cardiac muscles [50]. The parameter $k_{NCX}$ that describes the strain-dependence of the intracellular $Na^+$ concentration was adjusted to fit the slow response amplitude in human cardiac muscle samples [52]. The model was verified by comparison of the calculated frequency dependence of the isometric twitch amplitude (Fig 5) with data obtained on samples of human cardiac muscles [20].

## Reduction of previous models

To make the model as simple as possible computationally, while capable to simulate all major phenomena of the excitation-contraction coupling in human cardiac muscle, we neglected all $Ca^{2+}$ transmembrane currents except those through the L-type $Ca^{2+}$ channels, $I_{CaL}$, and the $Na^+$-$Ca^{2+}$ exchange ($I_{NCX}$). Compared to [3] $I_{CaL}$ was set as a function of membrane potential according to Eq (5) assuming that the fast opening of L-type channels is instantaneous while their slow closure can be neglected. The Eq (6) for $I_{NCX}$ was the same as in the model by ten Tusscher and Panfilov [3]. As subspace volume is very small ($\alpha_{ss} \ll 1$ in Eq (17)) in [3], it requires a very small time step for numerical integration. To avoid this problem, we employ Tikhonov theorem [47] and substituted differential Eq (17) with cubic algebraic Eq (19).

## Mechano-electrical and mechano-calcium feedbacks

Some modern models describe so called mechano-electical feedback, i.e. the changes in the AP durations upon changes in muscle length. Such changes could be caused by changes in $Ca^{2+}$ concentration in the cytosol accompanying $Ca^{2+}$ binding or release from Tn [14]. Some models include a description for the stretch activated channels [10, 16]. Some other models take into account the strain-dependence of myocardial electrical conductivity due to a change in geometry at constant specific conductivity [17, 60, 61]. A more detailed data-based model by de Oliveira et al. [19] describes a decrease in the AP propagation velocity upon muscle stretch caused by the strain-dependence of membrane capacitance and cell-to-cell conductivity caused

by a recruitment of caveolae to the sarcolemma as found by Pfeiffer et al. [27]. Although this process cannot be instantaneous, we assume, as a first approximation, the capacitance and the conductivity to be functions of current sarcomere length according to Eq (3). This equation was the only mechano-electrical feedback in our model, stretch-activated channels were neglected. The mechano-calcium feedback was described by two terms: the strain-dependence of the transition from the blocked to the activated state of the Tn-Tpm complex in Eqs (12) and (13) and the strain-dependence of $Na^+$ concentration in cytosol, $[Na^+]_i$, in Eq (6). Although an increase in $[Na^+]_i$ is possibly caused by strain-dependent changes in $Na^+$ exchange, and other mechanism(s) can also be involved to slow response to stretch [30], we used the simplest approximation that can describe this phenomenon.

## Novelty

The coupling of different models of cardiac mechanics and electrophysiology is a common approach for developing electromechanical models. We would like to highlight the following aspects of our model.

1. Our previous model of myocardium mechanics [25], which modification was used in the study, reproduced a large set of mechanical features observed in experiments performed on cardiac muscle samples or single cells, while being computationally simple. The model is specified by a system of ordinary differential equations for the kinetics of the interactions of the cardiomyocyte contractile and regulatory proteins and is similar, in this aspect, to some others model of myocardium mechanics [63–65]. However, the performance of our mechanical model is slightly better, and the set of experimental data reproduced by the model is larger than ones shown in the other model studies (see [25] for details). New modifications of the model described in Materials and Methods section were shown to fit some experimental data (force-calcium steady-state relationship) even better than the previous version of the model [25]. In addition, the modifications allowed us to improve computational efficiency of the model.

2. In order to reproduce the dependence of the twitch tension on the stimulations frequency, one should couple the mechanical block with the model electrophysiology. This includes the description for the variation of the AP and the specifications of the electromechanical AP-$Ca^{2+}$ coupling. We describe the AP with the simple two-variable phenomenological model of Aliev-Panfilov [21] that reproduce the general form of the AP time-course and the AP dependence on the stimulating frequency well and was used and approbated in numerous publications. Our description of AP-$Ca^{2+}$ coupling is mostly based on the one introduced in the model by ten Tusscher and Panfilov [3]. However, we simplified the equations from [3] significantly making sure that the simplified equations were still relevant to the physical processes that the equations describe. An important feature of our model, which was absent in [3], is the introduction of calcium-dependent $Ca^{2+}$ uptake into sarcoplasmic reticulum via Ca-dependent phosphorylation of protein(s) involved into the uptake. While there are several models of heart electromechanics in the literature, some of which use the models [21] or [3] for the electrophysiology description and in some cases include detailed descriptions of myocardium mechanics [9–11], we were not able to find the results of reproducing the dependence of the twitch force and its relaxation rate on stimulation frequency.

3. Another novel feature of our model is the mechano-electrical feedback provided by the strain-dependence of the cell membrane capacitance and cell conductivity taken from [19]. In numerical experiments reported here, the model variables did not depend on spatial

coordinate. However, preliminary results of the simulation of 2D muscle contraction demonstrate significant changes in the excitation-contraction waves caused by the strain-dependent electrophysiology. These results will be published elsewhere.

To summarize, we have not used some brand new approaches while developing our model, with exception for a couple of modifications of our previous model equations and some equations from the model [3]. However, we made an effort to combine and modify the models developed earlier to obtain new results that reproduce the important features of myocardium contraction with the least computational cost possible.

## Limitations

The main limitation of the model is the absence of a detailed description of the ion currents and their contribution to the AP. For this reason, the model cannot describe some important details of heart electrophysiology, and we could not validate patient- or species-specific parameters of the electrical part of the model. On the other hand, the Aliev-Panfilov model [21] used here reproduces many major phenomena of AP propagation in normal and diseased human heart including reentrant arrhythmias [22–24]. Also, only a few $Ca^{2+}$ flows and currents were included in the model while changes in concentrations of $Na^+$, $K^+$ and other ions were neglected. Therefore, any factors affecting these neglected processes cannot be reproduced by the model. Complicated cooperative processes involved in CIRC [39, 66] were not taken into account. Instead, a simple approach of ten Tusscher and Panfilov [3] was further simplified. We cannot exclude that our model is unable to reproduce changes in CICR at some heart diseases. The processes involved in the mechano-electrical and mechano-calcium feedbacks probably are not instantaneous. There is a delay between changes in sarcomere length and changes in the electrical and biochemical processes. The delay was not taken into account here. Nevertheless, the model reproduced the slow response to stretch of cardiac muscle that takes tens of seconds (Fig 3).

## Conclusions

The model suggested here combines simplified and modified version of previously described models [3, 21, 25]. The simplifications allowed us to make model more effective computationally due to the reduction of the number of differential equations and their stiffness compared to predecessor models. The modifications included the description of the mechano-electrical and mechano-calcium feedbacks and the dependence of $Ca^{2+}$ uptake into SR on $Ca^{2+}$-dependent phosphorylation of protein(s) involved into the uptake.

The model describes various mechanical experiments with cardiac muscle including load-dependent relaxation (Fig 7), steady-state contractions, nonstationary isometric twitches (Fig 2), both instantaneous and slow mechanical responses to muscle stretch (Figs 1 and 3), and the dependence of the twitch amplitude and duration on the interstimulus interval (Figs 4–6).

Because of the above-mentioned modification and simplifications of the model, the time step of the numerical integration by the explicit Euler method can be increased to 0.1–0.2 ms. As the Aliev-Panfilov [21] model and the mechanical model do not require small mesh, the model appears to be suitable for 3D simulation of heart electro-mechanics.

## Supporting information

**S1 File. Additional details of the model.** The rule of the choice of the solution of Eq (19) for $Ca^{2+}$ concentration in subspace, the 0D mechanical problem setup for muscle sample

contraction, and the results of the convergence test for the numerical method are specified. (PDF)

## Author Contributions

**Conceptualization:** Andrey Tsaturyan.

**Formal analysis:** Anna Osepyan.

**Funding acquisition:** Fyodor Syomin.

**Investigation:** Fyodor Syomin, Anna Osepyan, Andrey Tsaturyan.

**Methodology:** Andrey Tsaturyan.

**Software:** Fyodor Syomin, Andrey Tsaturyan.

**Supervision:** Andrey Tsaturyan.

**Validation:** Fyodor Syomin.

**Visualization:** Anna Osepyan.

**Writing – original draft:** Fyodor Syomin, Andrey Tsaturyan.

**Writing – review & editing:** Fyodor Syomin, Andrey Tsaturyan.

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
