## [Decision Letter · Decision Letter 0]

2 Jun 2021

PONE-D-21-14208

Computationally effective model of myocardial electromechanics for multiscale simulations

PLOS ONE

Dear Dr. Syomin,

Thank you for submitting your manuscript to PLOS ONE. After careful consideration, we feel that it has merit but does not fully meet PLOS ONE’s publication criteria as it currently stands. Therefore, we invite you to submit a revised version of the manuscript that addresses the points raised during the review process.

We look forward to receiving your revised manuscript.

Kind regards,

Alexander V Panfilov, PhD

Academic Editor

PLOS ONE

Journal Requirements:

Additional Editor Comments:

Please respond to the reviewers comments. In particular, please provide additional info on the numerical methodology used, on novelty of the approach, please extend the description of the model and account for additional comments on the results presentation mentioned by the reviewers.

Reviewers' comments:

Reviewer's Responses to Questions

**Comments to the Author**

1. Is the manuscript technically sound, and do the data support the conclusions?

Reviewer #1: Yes

Reviewer #2: Partly

2. Has the statistical analysis been performed appropriately and rigorously? 

Reviewer #1: N/A

Reviewer #2: N/A

3. Have the authors made all data underlying the findings in their manuscript fully available?

Reviewer #1: Yes

Reviewer #2: Yes

4. Is the manuscript presented in an intelligible fashion and written in standard English?

Reviewer #1: Yes

Reviewer #2: Yes

5. Review Comments to the Author

Reviewer #1: The paper introduces an original mathematical model of the electromechanical coupling (EMC) in the myocardium accounting for some mechano-electric and mechano-calcium feedback loops. The main message of the authors is to obtain computer tool efficient in terms of calculation speed, which will help solving complex problems of EMC modeling in the whole ventricle. For this purpose, the electrical block of the model has been simplified as much as possible. In particular, the AP generation is set by the Aliyev-Panfilov model equations, and only two currents contributing to the AP development are taken into account: slow L-type calcium current and Na+-Ca2+ exchange current.

The article presents results of the study in which the authors verified the ability of the model to simulate correctly the effects of myocardial mechanical activity and its calcium regulation. Both kinds of the effects pertain to the functions of the cardiomyocytes. In other words, the capabilitiy of the model to simulate such features of multicellular myocardial preparations as electrical and mechanical interactions between myocytes has not been utilized in this work as yet. This seems reasonable, since before modeling the effects of excitation-contraction coupling at the level of the entire ventricle it was necessary to confirm that significantly reduced description of the electrical compartment in the model did not prevent the correct simulation of the key effects at the cellular level. The slow responses of both active mechanical tension and calcium transients on the stretch of the muscle preparation and on the change of the pacing rate are especially important among such key effects. The authors have proved that the model adequately reproduces these effects.

The most impressive of these results is that concerning the force-frequency relationship. As far as the reviewer knows, this is the first electro-mechanical model that correctly reproduces and suggests explanation of simultaneous increases in the peak force and calcium transient amplitude in parallel with a decrease in the duration of twitches and calcium transients after a rise in the pacing rate.

The force and calcium signals in the model, as well as the effects listed above, were validated by comparison with published data obtained on the human myocardium. Of course, very simplified description of the action potential generation used in the model does not imply its validation, since such action potential cannot be attributed not only to the human myocardium, but to any other species as well. This does not seem to be a drawback of the work, but rather its specific feature. Nevertheless, this circumstance would be right to point out in the text somewhere among the limitations of the model.

In general, the paper is appropriate for publication in PLOS ONE after a few minor revisions listed below, which will make some of its aspects more understandable to readers.

MINOR COMMENTS

1. Either at the end of the model description section or at the beginning of the section "Results", it would be helpful to state more clearly the 0D-conditions applied to the twitch simulations you ran in the 3D model. In particular, it should be clarified that for the correct simulation of the single cardiomyocyte behavior, you eliminated the factors of electrical and mechanical interaction between the cells in the 3D media, which was done by the identity of the cells' parameters and by their simultaneous stimulation.

2. The sentence

«Following [19] we assumed that the cell membrane capacitance C normalized for its value at =1 and d1 depend on the axial strain ... as follows: .... »

on page 9 contains term d1 that previously arose only once within formula (2) on page 5 and was defined there in quite indirect way. This is very inconvenient for the readers. I'd suggest to clarify right after this phrase on page 9 that d1 is the longitudinal component of the conductivity tensor D.

3. It seems to be reasonable to add informative notations before the formal symbols A1 and A2 in section "Results" and in the respective figure legends. For example, such notations as «concentration of the calcium-troponin complexes in the overlap zone (A1)», «… in the non-overlap zone (A2)» would help readers to understand your results easier.

4. Fig. 2 legend: «The time courses of model variables ... Ta (normalized for its maximal value)». It makes sense to explain here more precisely what maximum is meant. Probably you mean the absolute maximal value calculated for the steady-state conditions at saturating calcium concentration. However this is not explained. Meanwhile, by default the quoted phrase is perceived as normalizing to the force amplitude just of the presented twitch. Thus it becomes unclear why this amplitude in the figure turned out not equal to 1.

5. The conditions of the numerical experiment presented in Fig. 5B, and their differences from the experimental conditions corresponding to Fig. 5A are described very unclear. It should be clarified that the regular pacing with the rate of 1 Hz was stopped followed by one extrasystolic potentiation (for points corresponding to the intervals shorter than 1 s on panel 5B) or by one post-pause potentiation (for points corresponding to the intervals longer than 1 s on this panel).

Reviewer #2: The manuscript is concerned with the development of a computationally efficient approach to cardiac electromechanics. For this purpose, the authors combine the previously developed electrophysiology models of Aliev and Panfilov (1996), ten Tusscher & Panfilov (2006) and the electromechanical model recently proposed by Syomin and Tsaturyan (2017). The authors extend the Aliev-Panfilov model to incorporate different types of calcium transients using the expressions suggested in the tenTusscher-Panfilov model. Furthermore, the constitutive equation describing the evolution of active stress accounts for various physiologically observed phenomena that include the effect of velocity on stress and stiffness, responses to strain and load changes, load-dependent relaxation etc. Moreover, the authors propose some simplifications to make the governing differential equations less stiff. The computational results presented by the authors illustrate the capabilities of the proposed approach.

Remarks:

1. It may be more appropriate to use the adjective "efficient" than "effective" in the title.

2. The authors are strongly recommended to the clearly highlight the novel aspects of the proposed approach compared both to their own model (Syomin and Tsaturyan (2017)) and to the similar models suggested in literature.

3. In the numerical setting, the authors claim that the explicit Euler method gives convergent results for the time step dt=0.1 ms. This should be shown in a set of plots generated by convergence analyses reporting transients of some of the model outcomes obtained for different dt values.

4. Although the governing equation of electrophysiology (1) is a transient PDE, in the numerical analyses, the authors seem to solve solely a set of ODEs. Therefore, it is unclear whether the claims/findings concerning the reduction of stiffness and convergence with respect to the time step hold also for PDEs including the conservation of linear momentum.

5. In the manuscript, some abbreviations such as TnC, Tpm-Tn remain undefined. The meaning of all the abbreviations mush be explained.

In general, the reviewer has enjoyed reading the manuscript . Nevertheless, the manuscript lacks strength in some aspects, some of which are mentioned above. Therefore, the authors are urged to thoroughly address these concerns if they revise their work.

6. PLOS authors have the option to publish the peer review history of their article (what does this mean?). If published, this will include your full peer review and any attached files.

Reviewer #1: **Yes: **Leonid B. Katsnelson

Reviewer #2: No

---

## [Author Response · Author response to Decision Letter 0]

17 Jun 2021

Dear prof. Panfilov,

We are very grateful to you and both reviewers for careful reading and evaluation of our manuscript and very useful comments and suggestions. We have prepared revised version of the manuscript with changes and additions suggested by the reviewers. The point to point responses to the reviewers’ comments are listed below.

Review Comments to the Author

Reviewer #1: 

Of course, very simplified description of the action potential generation used in the model does not imply its validation, since such action potential cannot be attributed not only to the human myocardium, but to any other species as well. This does not seem to be a drawback of the work, but rather its specific feature.

Nevertheless, this circumstance would be right to point out in the text somewhere among the limitations of the model.

We agree with this comment and added a sentence concerning the action potential modelling and the validation of the electrophysiological block of the model.

MINOR COMMENTS

1. Either at the end of the model description section or at the beginning of the section "Results", it would be helpful to state more clearly the 0D-conditions applied to the twitch simulations you ran in the 3D model. In particular, it should be clarified that for the correct simulation of the single cardiomyocyte behavior, you eliminated the factors of electrical and mechanical interaction between the cells in the 3D media, which was done by the identity of the cells' parameters and by their simultaneous stimulation.

A sentence explaining the details of the simulations is added to the revised manuscript.

2. The sentence «Following [19] we assumed that the cell membrane capacitance C normalized for its value at =1 and d1 depend on the axial strain ... as follows: .... »

on page 9 contains term d1 that previously arose only once within formula (2) on page 5 and was defined there in quite indirect way. This is very inconvenient for the readers. I'd suggest to clarify right after this phrase on page 9 that d1 is the longitudinal component of the conductivity tensor D.

Additional explanations were added to the revised manuscript as suggested.

3. It seems to be reasonable to add informative notations before the formal symbols A1 and A2 in section "Results" and in the respective figure legends. For example, such notations as «concentration of the calcium-troponin complexes in the overlap zone (A1)», «… in the non-overlap zone (A2)» would help readers to understand your results easier.

The physical meaning of the variables A1 and A2 was added to the revised manuscript as suggested.

4. Fig. 2 legend: «The time courses of model variables ... Ta (normalized for its maximal value)». It makes sense to explain here more precisely what maximum is meant. Probably you mean the absolute maximal value calculated for the steady-state conditions at saturating calcium concentration. However this is not explained. Meanwhile, by default the quoted phrase is perceived as normalizing to the force amplitude just of the presented twitch. Thus it becomes unclear why this amplitude in the figure turned out not equal to 1.

The details for the tension normalizations are added to the Figure 2 legend in the revised manuscript.

5. The conditions of the numerical experiment presented in Fig. 5B, and their differences from the experimental conditions corresponding to Fig. 5A are described very unclear. It should be clarified that the regular pacing with the rate of 1 Hz was stopped followed by one extrasystolic potentiation (for points corresponding to the intervals shorter than 1 s on panel 5B) or by one post-pause potentiation (for points corresponding to the intervals longer than 1 s on this panel).

Explanations of the conditions of the numerical experiments shown in Fig. 5 were added to the revised manuscript as suggested.

Reviewer #2: 

1. It may be more appropriate to use the adjective "efficient" than "effective" in the title.

The title of the revised manuscript was changes as suggested.

2. The authors are strongly recommended to the clearly highlight the novel aspects of the proposed approach compared both to their own model (Syomin and Tsaturyan (2017) and to the similar models suggested in literature.

A new subsection summarizing the novelty of our approach and results was added into the discussion section.

3. In the numerical setting, the authors claim that the explicit Euler method gives convergent results for the time step dt=0.1 ms. This should be shown in a set of plots generated by convergence analyses reporting transients of some of the model outcomes obtained for different dt values.

We are sorry for the mistyping in the first para of Results section that could be misleading: the time-steps tested for the convergence analysis were 0.2, 0.1, and 0.05 ms. The results of the convergence analysis are provided in S1 Fig. 3 of the Supporting information file of the revised manuscript.

4. Although the governing equation of electrophysiology (1) is a transient PDE, in the numerical analyses, the authors seem to solve solely a set of ODEs. Therefore, it is unclear whether the claims/findings concerning the reduction of stiffness and convergence with respect to the time step hold also for PDEs including the conservation of linear momentum.

In the manuscript revised, we indeed solved solely a system of ODEs. However, we also tested the model for the simulation of 2D processes. In this case, we implemented the finite element method for finding transmembrane potential and mechanical displacements in the nodal points and numerically solved ODEs for all other kinetic variables using the explicit Euler method. The time-step of 0.1 ms provided good convergence, as it did for the solution of the ODE system for the 0D simulations presented in the manuscript and Supporting Information. The details of the 2D simulations will be published elsewhere.

5. In the manuscript, some abbreviations such as TnC, Tpm-Tn remain undefined. The meaning of all the abbreviations mush be explained.

All abbreviations are introduced in the revised manuscript as suggested.

Yours, sincerely

Fyodor Syomin, PhD

Research Scientist

Department of Biomechanics

Institute of Mechanics

Lomonosov Moscow State University

---

## [Decision Letter · Decision Letter 1]

9 Jul 2021

Computationally efficient model of myocardial electromechanics for multiscale simulations

PONE-D-21-14208R1

Dear Dr. Syomin,

We’re pleased to inform you that your manuscript has been judged scientifically suitable for publication and will be formally accepted for publication once it meets all outstanding technical requirements.

Kind regards,

Alexander V Panfilov, PhD

Section Editor

PLOS ONE

Additional Editor Comments (optional):

Reviewers' comments:

Reviewer's Responses to Questions

**Comments to the Author**

1. If the authors have adequately addressed your comments raised in a previous round of review and you feel that this manuscript is now acceptable for publication, you may indicate that here to bypass the “Comments to the Author” section, enter your conflict of interest statement in the “Confidential to Editor” section, and submit your "Accept" recommendation.

Reviewer #1: (No Response)

Reviewer #2: All comments have been addressed

2. Is the manuscript technically sound, and do the data support the conclusions?

Reviewer #1: (No Response)

Reviewer #2: Yes

3. Has the statistical analysis been performed appropriately and rigorously? 

Reviewer #1: (No Response)

Reviewer #2: N/A

4. Have the authors made all data underlying the findings in their manuscript fully available?

Reviewer #1: (No Response)

Reviewer #2: Yes

5. Is the manuscript presented in an intelligible fashion and written in standard English?

Reviewer #1: (No Response)

Reviewer #2: Yes

6. Review Comments to the Author

Reviewer #1: (No Response)

Reviewer #2: The reviewer is satisfied with the revisions conducted by the authors. Therefore, the manuscript is recommended for publication in the PONE.

7. PLOS authors have the option to publish the peer review history of their article (what does this mean?). If published, this will include your full peer review and any attached files.

Reviewer #1: No

Reviewer #2: No

---

## [Editor Report · Acceptance letter]

14 Jul 2021

PONE-D-21-14208R1 

Computationally efficient model of myocardial electromechanics for multiscale simulations 

Dear Dr. Syomin:

I'm pleased to inform you that your manuscript has been deemed suitable for publication in PLOS ONE. Congratulations! Your manuscript is now with our production department. 

Kind regards, 

on behalf of

Prof. Alexander V Panfilov 

Section Editor

PLOS ONE